# The Role and Clinical Interest of Extracellular Vesicles in Pregnancy and Ovarian Cancer

**DOI:** 10.3390/biomedicines9091257

**Published:** 2021-09-18

**Authors:** Nazanin Yeganeh Kazemi, Benoìt Gendrot, Ekaterine Berishvili, Svetomir N. Markovic, Marie Cohen

**Affiliations:** 1Mayo Clinic Medical Scientist Training Program, Mayo Clinic, Rochester, MN 55905, USA; kazemi.nazanin@mayo.edu; 2Department of Pediatrics, Gynecology, Obstetrics, Faculty of Medicine, University of Geneva, 1206 Geneva, Switzerland; benoit.gendrot@etu.u-paris.fr; 3Translational Research Center in Hematologic Oncology, Faculty of Medicine, University of Geneva, 1206 Geneva, Switzerland; 4Laboratory of Tissue Engineering and Organ Regeneration, Department of Surgery, University of Geneva, 1206 Geneva, Switzerland; ekaterine.berishvili@unige.ch; 5Mayo Clinic Department of Oncology, Mayo Clinic, Rochester, MN 55905, USA; markovic.svetomir@mayo.edu

**Keywords:** extracellular vesicle, exosome, syncytial knot, pregnancy, placenta, ovarian cancer, invasion, immune modulation, angiogenesis, preeclampsia

## Abstract

Ovarian cancer and pregnancy are two states in which the host immune system is exposed to novel antigens. Indeed, both the tumor and placenta must invade tissues, remodel vasculature to establish a robust blood supply, and evade detection by the immune system. Interestingly, tumor and placenta tissue use similar mechanisms to induce these necessary changes. One mediator is emerging as a key player in invasion, vascular remodeling, and immune evasion: extracellular vesicles (EVs). Many studies have identified EVs as a key mediator of cell-to-cell communication. Specifically, the cargo carried by EVs, which includes proteins, nucleic acids, and lipids, can interact with cells to induce changes in the target cell ranging from gene expression to migration and metabolism. EVs can promote cell division and tissue invasion, immunosuppression, and angiogenesis which are essential for both cancer and pregnancy. In this review, we examine the role of EVs in ovarian cancer metastasis, chemoresistance, and immune modulation. We then focus on the role of EVs in pregnancy with special attention on the vascular remodeling and regulation of the maternal immune system. Lastly, we discuss the clinical utility of EVs as markers and therapeutics for ovarian cancer and pre-eclampsia.

## 1. Introduction

Cancer and pregnancy represent two physiologic states in which immunity must be modulated for the growth of new tissue expressing novel antigens. Thus, tumor and placenta tissue have both evolved mechanisms to evade host immunity [1,2]. The importance of immunosuppression in tumor growth has been recognized for several decades leading to the development of immune checkpoint inhibitors which continue to demonstrate efficacy in several tumor types [3]. Cancer cells often carry a high mutation burden which can contribute to the expression of tumor-associated antigens that the adaptive immune system can recognize and target [4]. Just as malignant cells can express novel antigens, fetal and placenta tissue are only haploidentical to the mother with the capacity to express antigens not yet encountered by the maternal immune system. In pregnancies involving gestational carriers, the carrier shares potentially no genetic background with the fetus but can carry the fetus to term. If we consider the fetus to be a non-HLA matched tissue graft in this scenario, we must question how the carrier’s immune system does not reject the fetus and allows it to grow to term. A major contributor to this tolerance is the placenta which inhibits the maternal immune responses much in the same way that tumors suppress host immunity [5].

Malignant and placenta cells both proliferate rapidly, invade tissues, establish robust blood supplies, and create a microenvironment which renders immune cells, otherwise capable of recognizing foreign antigens expressed by these tissues, ineffective at mediating a response [2]. Like tumor cells, trophoblasts of the placenta upregulate telomerase and survivin expression to increase cell division while inhibiting apoptosis [6,7]. To invade surrounding tissues or the uterus, tumor cells and trophoblasts decrease E-cadherin expression and secrete matrix metalloproteases to mediate cell migration. Molecular pathways involved in tissue invasion are also shared between tumor cells and trophoblasts including the JAK-STAT pathway, Rho-associated kinase, MAPKs, PI3K, and SMAD family proteins [8,9]. Furthermore, both tissues rely on VEGF and mTOR pathways for robust neoangiogenesis [10,11].

Many studies have demonstrated that tumor and placenta tissue use the same mechanisms to suppress host immunity [2]. Approximately 40% of cells in the decidua are innate immune cells with uterine NK (uNK) cells making up the majority of immune cells at the maternal-fetal interface [12]. In contrast to peripheral NK cells, uNK cells do not express CD16, the FcRγIIIA receptor, meaning that they do not participate in antibody-dependent cell-mediated cytotoxicity [13,14]. In fact, uNK cells serve a modulatory rather than a cytotoxic role in pregnancy by inducing differentiation of dendritic cells to a tolerogenic phenotype [13]. Interestingly, tumor-associated NK cells share these characteristics of uNK cells allowing tumor cells to persist despite low levels of major histocompatibility complex I (MHCI) expression [15]. Regulatory T cells (T_regs_) are another integral part of immunosuppression both within the uterus and systemically during pregnancy. Decreases in T_reg_ numbers during pregnancy can be associated with spontaneous abortion and pre-eclampsia which are associated with perinatal and maternal mortality [16,17,18,19]. T_regs_ have long been identified as drivers of immunosuppression in cancer by inhibiting antigen-specific responses and inflammation [20]. In pregnancy, T cell function is skewed toward a T_reg_ and Th2 dominated phenotype by the presentation of antigens in a tolerogenic context [21]. For example, dendritic cells (DCs) are a CD83^+^ Th2-promoting phenotype which induce angiogenesis and tolerance in pregnancy [22]. Deficiency of DCs results in resorption of fetal tissue in murine models even in syngeneic pregnancy in which neoantigens are absent [23]. It is known that DCs in cancer have a similar phenotype induced by the secretion of immunosuppressive mediators from the tumor [24]. Tumor and placenta cells also downregulate the expression of HLA-A, B, and C alleles thus decreasing antigen presentation altogether [25,26]. This ensures that cells expressing novel antigens are not detected and targeted for apoptosis while modulation of NK cell activity ensures that cells with low HLA expression levels are not killed. Instead of *MHCI* genes, tumor and placenta tissue express HLA-G which interacts with inhibitory receptors on NK cells and cytotoxic CD8^+^ T cells to thwart apoptosis [27,28,29]. While necessary for a successful pregnancy, HLA-G expression is associated with poor outcomes in many different cancer types [30,31]. Other well-known inhibitory signals expressed by tumor and placenta cells include FASL, TRAIL, and B7H1 which can induce lymphocyte death or anergy. Secretion of inhibitory mediators such as interleukin 10 (IL-10), macrophage migration inhibitory factor (MIF), and indoleamine 2,3-dioxygenase (IDO) also play a role in immunosuppression in these two tissues. Through the expression of these surface signals and secreted mediators, tumor and placenta tissue can inhibit immune cells which may enter the tissue and respond to new antigens [2].

A newly recognized mediator of angiogenesis, invasion, and immunosuppression in both pregnancy and cancer is extracellular vesicles (EVs) [32]. EVs are secreted from all cells of the body and can be found in almost all bodily fluids. These particles can be divided into different classes based on their compartment of origin within the cell, size, surface marker expression, and cargo (Figure 1). Large EVs range from 500–2000 nm in diameter and include apoptotic bodies from dying cells and syncytial nuclear aggregates (syncytial knots) from cytotrophoblast syncytialization into syncytiotrophoblasts. These particles contain chromatin and organelles from the cell of origin [33]. Apoptotic bodies express phosphatidylserine on their surfaces [34]. Medium EVs are microvesicles which are derived by budding from the cell membrane. Microvesicles range in size from 50–1000 nm in diameter and can express surface markers specific to the cell of origin. However, there are several markers that are used to identify microvesicles regardless of their origin cell including integrins, selectins, and CD40. Microvesicles contain proteins and nucleic acids including mRNA, miRNA, and other non-coding RNAs. Lastly, exosomes are the smallest EVs ranging in size from 40–200 nm in diameter and forming from endosomes which give rise to multivesicular bodies (MVBs). MVBs are trafficked and sorted through the endosomal sorting complexes required for transport (ESCRT) pathway with some being sent to the lysosome for degradation while the contents of others, including exosomes, are secreted from the cell. Again, the cell of origin determines any specific cargo found within exosomes; however, their origin from MVBs and trafficking through ESCRT introduces several markers that are shared regardless of cell of origin including Alix, TSG101, tetraspanins (CD81, CD63, and CD9), and flotillin-1. Exosomes and microvesicles transport similar cargo [35].

While apoptotic bodies mediate phagocytosis of dead cell debris [36], recent studies have demonstrated the role of microvesicles and exosomes in cell-to-cell communication [37]. Depending on their cargo, these particles can modulate recipient host activity. Specifically, EVs shed by cancer cells can express markers which modulate immunity and even interfere with cancer treatment including PD-L1 [38]. Placenta-derived EVs have also been demonstrated to affect maternal immunity throughout each trimester [39]. In this review, we highlight the role of EVs in angiogenesis, tissue invasion, and immunosuppression in ovarian cancer (OC) and pregnancy with a specific focus on how particles shed from these tissues interact with the immune system.

## 2. EVs in Ovarian Cancer

### 2.1. EVs from Ovarian Cancer Contribute to Metastasis, Establishment of the Premetastatic Niche and Chemoresistance

Ovarian cancer is the seventh most common cause of cancer death in women and the leading cause of mortality from gynecologic cancer [40]. Tumors arising from the ovary can be divided based on the cell type. The majority (90%) of ovarian tumors are derived from epithelial cells. The remaining 10% which are not of epithelial origin are less invasive. The majority of epithelial ovarian cancers are serous ovarian cancers with endometroid, clear cell, mucinous and unspecified tumors making up the remainder [40]. Non-specific signs and symptoms including pelvic or abdominal pain, early satiety and increased abdominal size delay diagnosis until later stages of disease.

Anatomically, tumors arising from the ovary are well-positioned to disseminate throughout the peritoneum causing significant disease burden. Seeding of the peritoneum and spread through ascitic fluid occurs and contributes to the poor prognosis of ovarian cancer [41]. Therefore, it is imperative to understand the mechanisms of peritoneal metastasis to develop strategies for early intervention. Recent studies demonstrate that EVs released from ovarian cancer cells may facilitate the escaping of ovarian cancer cells. Indeed, vesicles from metastatic type I epithelial ovarian cancer can induce apoptosis of mesothelial cells which normally create a barrier between the ovaries and abdominal cavity [42]. EVs derived from ovarian cell lines and primary ovarian tumors deliver mRNA encoding matrix metallopeptidase 1 (MMP1) which can degrade extracellular matrix components and induce activation of Caspase 1 thus triggering apoptosis in receiving cells [43] (Figure 2). This demonstrates that EVs can deliver cargo which induces apoptosis in barrier tissues such as the mesothelium enhancing metastasis of ovarian cancer through the ascitic fluid. Ovarian tumor-derived EVs can also transfer MMP2 and MMP9 proteins to recipient cells thereby mediating the degradation of extracellular matrix components to allow for tumor cell migration [44].

Other studies have demonstrated that epithelial ovarian tumor EVs can transfer CD44 to mesothelial cells thus inducing the epithelial-mesenchymal transition, downregulating E-cadherin, and inducing MMP9 expression which promotes ovarian cancer invasion and metastasis through degradation of the extracellular matrix [45]. Transfer of LIN28 by tumor EVs can induce a metastatic phenotype in previously non-metastatic recipient cells and induce the epithelial-mesenchymal transition in co-cultured cells through the expression of required genes including *NOTCH1, WNT5A, NODAL, ZEB1,* and *SNAI2* [46]. Additionally, claudin-4 overexpression in ovarian tumor-derived EVs can alter paracellular permeability to facilitate metastasis [47].

In addition to their direct involvement in metastasis, ovarian cancer EVs are also involved in the establishment of the premetastatic niche which provides a favorable microenvironment for initial metastatic cells. Ovarian tumor EVs can convert healthy fibroblasts to further differentiated cancer-associated fibroblasts (CAFs) which establish an environment that favors angiogenesis, malignant cell invasion, and tumor growth. CAFs also secrete factors which remodel the tumor stroma to inhibit immune responses [48,49] (Figure 2). Induction of fibroblast differentiation to the CAF phenotype, in turn, promotes tumor cell migration. Specifically, CAFs release exosomes containing TGF-β which triggers the SMAD signaling cascade leading to increased cell migration [50]. Additionally, several miRNAs released by ovarian tumor cells, and packaged within exosomes, contribute to the establishment of the tumor microenvironment. These miRNAs induce tumor cell invasion, immunotolerance, and mesothelial cell clearance [51]. It has been demonstrated that ascitic fluid from ovarian cancer patients contains EVs which may promote tumor cell migration necessary in metastasis by transferring molecules such as CD24 and EpCAM to recipient cells which are associated with increased tumor invasiveness [52,53].

In addition to late diagnosis and early metastasis, another feature of ovarian cancer which contributes to poor prognosis is resistance to chemotherapy [54]. Pathways to chemotherapy resistance include survival and proliferation of malignant cells with mutated genes affecting drug targets, drug metabolism, and efflux pumps. EVs have been demonstrated to play a role in the development of this resistance. Specifically, through the delivery of miRNAs, EVs can induce the expression of detox enzymes and downregulate chemotherapy targets [55].

Other studies have demonstrated that treatment of ovarian cancer cells with cisplatin can induce the release of EVs which act on surrounding cells to increase invasion and drug resistance [56] (Figure 2). Known as the bystander effect, this phenomenon refers to the release of EVs from cells under potentially cytotoxic stress such as heat or chemotherapy. The EVs released from stressed cells can induce a stressed state in cells which have not been exposed to the stressor [57]. Thus, EVs from ovarian cancer cells treated with cisplatin, but not untreated controls, are able to alter the phosphorylation state of key signaling proteins including downregulation of CREB, ERK2, and TOR phosphorylation with upregulation of JNK, p53, and p38 phosphorylation. These changes in phosphorylation state are associated with chemotherapy resistance. Specifically, p38 activation has been previously demonstrated to play a role in chemoresistance. This study also demonstrates how inhibition of EV uptake can sensitize cells to cisplatin treatment [56]. Thus, EVs from tumor cells exposed to chemotherapy may be a missing link in the induction of resistance. Furthermore, studies using ovarian cancer spheroids to model the effects of cancer stem cells demonstrate that EVs released after treatment with cisplatin can increase the migration of bone marrow mesenchymal stem cells (MSCs). Subsequently, cisplatin induces secretion of IL-6, IL-8, and VEGFA from MSCs which induces angiogenesis and increases migration of previously non-invasive cancer cells [58].

In response to carboplatin therapy, ovarian cancer cells release EVs with miRNA cargo that mediates chemoresistance and increases malignant cell viability. Specifically, miR-21-5p alters the metabolism of receiving cells to increase glycolysis and the expression of adenosine triphosphate (ATP)-binding cassette transporter (ABCB6) and detoxification enzyme (GSHB). miR-21-3p and miR-891-5p increase the expression of DNA mismatch repair enzymes and MYC targets thus helping to overcome the deleterious effects of carboplatin therapy and promote cell cycle progression despite treatment [55].

Lastly, in determining the role of ovarian tumor-derived EVs in angiogenesis, studies have demonstrated that EVs from ovarian cancer cells transfer miR-141-3p to endothelial cells. This miRNA upregulates the JAK-STAT3 pathway through decreased expression of cytokine-inducible suppressors of cytokine signaling (SOCS)-5. Furthermore, miR-141-3p has potential roles in the upregulation of VEGFR-2 expression on endothelial cells which induces endothelial cell migration [59].

### 2.2. Ovarian Cancer EVs and Immunology

Inhibition of the immune response is key to tumor progression and metastasis in all cancer types. Immunosuppression in cancer is evident not just within the tumor microenvironment but also systemically [60]. Specifically, ovarian cancer progression involves tumor-associated macrophages (TAMs) which stimulate the differentiation and activity of T_regs_ to establish a tumor microenvironment that effectively evades the immune system. Peritoneal tissue from patients with metastatic ovarian cancer demonstrates a higher level of T_regs_ than Th17 cells compared to peritoneal tissue from patients with benign ovarian tumors. In fact, the T_reg_/Th17 ratio can be used as a prognostic factor for overall survival in patients with epithelial ovarian cancer [61]. Exosomes have a unique role to play in this imbalance. Specifically, TAM-derived exosomes can transfer miRNAs (miR-29a-3p and miR-21-5p) to helper T cells which inhibits STAT3 signaling causing an imbalance between T_regs_ and Th17 cells which favors T_reg_ function [61]. Ovarian tumor cell lines exposed to hypoxic stress secrete exosomes containing miRNAs that promote STAT3 phosphorylation which induces TAMs to assume a non-inflammatory M2 phenotype favoring tissue repair and fibrosis [62] instead of antigen presentation.

In addition to tipping this balance between inflammation and immunosuppression, ovarian tumor-derived EVs induce expansion of T_regs_ while increasing their inhibitory activity and survival. This study demonstrated that microvesicles in ovarian cancer ascitic fluid contain IL-10, TGF-β, and FasL. Co-culture of these microvesicles with CD4^+^ T cells induces their differentiation into functional CD4^+^, CD25^+^, FOXP3^+^ T_regs_. Thus, whether through action on macrophages or direct interaction with T cells, tumor-derived EVs are able to promote T_reg_ expansion, function, and survival [63].

In addition to their modulation of innate immunity through macrophage function, tumor-derived EVs also act on monocytes. Exosomes from ovarian cancer cells can activate Toll Like Receptor (TLR) signaling in monocytes leading to IL-6 production. Activation of STAT3 by IL-6 contributes to the immunosuppressive phenotype of the tumor microenvironment [64]. At the interface between the adaptive and innate arms of the immune system are professional antigen presenting cells (APCs) that can present antigen while providing costimulatory signals to antigen-specific T cells. The most effective APCs are dendritic cells which can pick up antigens and present them to T cells in the draining lymph node. Exosomes carrying Arg1 in the ascites and plasma of ovarian cancer patients can also drain to the lymph nodes where they are taken up by dendritic cells and inhibit antigen-specific T cell activation [65].

Additionally, ovarian tumor EVs can inhibit NK cell cytotoxicity. Studies demonstrate that EVs from the malignant ascitic fluid are taken up by NK cells and increase tumor growth in mouse models. Thus, there is evidence to suggest that ovarian tumor-derived EVs can prevent tumor cell death through inhibition of NK cells which would normally detect the decreased MHCI expression on tumor cells as a danger signal [66]. Ovarian tumor EVs can also have direct cytotoxic effects on T cells thus evading another arm of immune-mediated cell death. Exosomes carrying FasL not only downregulate the expression of T cell receptors (TCRs) but activate T cell apoptosis as well. This acts as an effective mechanism by which tumor cells can suppress antigen-specific responses without directly interacting with T cells and risking detection. Additionally, exosomes from the ascitic fluid of ovarian cancer patients can suppress TCR machinery by inhibiting CD3-zeta and JAK3. Thus, TCRs are rendered non-functional and antigen-specific responses are inhibited by targeting key signal transduction machinery [67].

Exosomes from ovarian tumors can have a direct effect on T cell activation not only by inhibiting signal transduction, but also by preventing the translocation of NFκB and nuclear factor of activated T-cells (NFAT) into the nucleus which is the downstream effect of TCR signaling required for T cell activation [68]. These transcription factors increase the production and secretion of cytokines which act in an autocrine and paracrine manner to increase T cell activity and survival. For T cells to become activated, these key transcription factors must enter the nucleus where they induce cytokine production and T cell proliferation. Inhibition of T cell activation seems to be mediated by the expression of phosphatidylserine on the surface of ovarian tumor-derived EVs [68,69]. Studies have demonstrated that gangliosides on the surface of exosomes can also prevent T cell activation. Specifically, ganglioside GD3 on the surface of exosomes isolated from malignant ascitic fluid inhibits T cell activation through the TCR [70].

Lastly, in addition to the effects of EVs on specific immune cell populations, co-culture of ascites-derived EVs from ovarian cancer patients with peripheral lymphocytes, hematopoietic stem cells, and DCs demonstrates that these EVs are able to induce apoptosis in a variety of peripheral immune cells. This demonstrates that regardless of the mechanism of action on specific activation and proliferation pathways, tumor-derived EVs can induce immune cell death [71]. In addition to apoptosis, EVs from peritoneal fluid of patients with ovarian cancer ascitic fluid increases the expression of immunosuppressive genes in lymphocytes compared to exosomes from ascitic fluid of patients with ovarian cysts. These broad effects on immune cell activation can ensure that any lymphocytes in the tumor microenvironment can become skewed toward an inhibitory phenotype [72].

## 3. Placenta-Derived EVs in Pregnancy

Extracellular vesicles can be detected in maternal circulation during pregnancy, increasing in concentration as pregnancy progresses and in gestational complications [32]. Current studies demonstrate that EVs from the placenta and fetus can transport cargo to recipient maternal cells thus playing a necessary role in different gestational processes including trophoblast migration, placenta implantation, and tissue/vascular invasion required for placentation [73]. EVs provide a mechanism of communication between maternal and fetal tissues. For example, epithelial cells from the endometrium treated with estrogen and progesterone secrete EVs containing proteins involved in implantation such as fibulin1, laminin-α5, and type XV collagen [74]. Upon uptake of these endometrial EVs, trophoblast cells demonstrate increased adhesion through activation of the focal adhesion kinase (FAK) pathway. Similar to tumor-derived EVs, miRNAs also play a role in cell adhesion and invasion. For example, miR-30d from endometrium can increase the expression of Itgb3, Itga7, and Cdh5 which promote embryo adhesion in mouse models [32].

Within the placenta, three trophoblastic cell types can be found: extravillous cytotrophoblasts (EVTs), villous cytotrophoblasts (vCTBs), and syncytiotrophoblasts (STBs). EVTs are responsible for the invasion of the maternal decidua for implantation and establishment of vascular connections between maternal and fetal tissue. vCTBs are precursor cells to STBs which line placental villi. vCTBs fuse and differentiate to form multinuclear STBs which form the outer surface of the fetal placenta and have specialized functions such as hormone secretion, transport of nutrients and gases, removal of waste products, and maintenance of immune tolerance [75]. STBs are responsible for the shedding of syncytial knots and represent the primary source of placenta-derived EVs found in maternal circulation, thus playing a key role in feto-maternal communication [73].

### 3.1. Role of Placenta-Derived EVs in Vascular Remodeling

As in cancer, angiogenesis and vascular remodeling are crucial parts of implantation and placenta growth. Specifically, uterine spiral artery remodeling is required for the establishment of a route between maternal and fetal circulation. This requires the recruitment of vascular smooth muscle cells to the tissue. Studies have demonstrated that exosomes from EVT cell lines are involved in vascular remodeling through the delivery of MMPs to vascular smooth muscle and endothelial cells [76]. EVs from the placenta, including from EVTs, vCTBs, STBs, and mesenchymal stem cells, promote vascular cell migration necessary for angiogenesis [77]. Lastly, exosomes from EVTs can increase TNFα production by human umbilical vein endothelial cells (HUVECs) when the source cells are grown under conditions with low oxygen tension. Induction of TNFα was associated with a miRNA profile specific to EVT grown in hypoxic conditions and caused a decrease in endothelial cell migration [78].

### 3.2. Immunomodulatory Activity of EVs in Pregnancy

Similar to tumor tissue, the placenta also sheds EVs which can inhibit inflammation and recognition of fetal tissue by maternal immune cells. While cancer represents a globally suppressed immune state, the inflammatory landscape in pregnancy depends on gestational age [79,80]. Specifically, the first trimester can be associated with inflammation from implantation of the placenta and the establishment of the fetal-maternal interface. After this and up to parturition, however, the maternal immune system is suppressed to allow for the continued growth of the fetus, which expresses antigens encoded by both maternal and paternal genes [81]. A potential role for EVs in the first trimester has been demonstrated by examining the proteome of EVs from over 50 first-trimester placentas [82]. This study isolated EVs from placenta explants and demonstrated an increase in the expression of proteins involved in vesicular uptake and inflammation including annexin V, calreticulin, complement proteins, and minor histocompatibility antigens. Additionally, other studies have demonstrated the expression of FasL on first-trimester placenta EVs [83,84].

STB-derived EVs dampen immunologic responses which can promote cell death. For example, they express ligands for inhibitory receptors on NK cells including MIC-A/B and UL-16 binding proteins. Thus, while STBs do not express MHCI which allows the placenta to evade detection by maternal T cells recognizing paternal and fetal antigens, they also evade NK cell death which classically targets cells that have downregulated MHCI expression [85]. The HLA-null nature of STBs withstanding, these cells can express minor histocompatibility antigens encoded by paternal genes. Studies have demonstrated that two such antigens DDX3Y and HA-1 are found on the surface of syncytial nuclear aggregates where they may induce differentiation of antigen-specific T_regs_. Through this mechanism, EVs play a role in tolerizing maternal immune cells to fetal-specific antigens [86].

As in cancer, a hallmark of immunosuppression in pregnancy is inhibition of the adaptive immune response. FasL on exosomes released from first-trimester placentas can induce apoptosis of T cells [83]. Further studies have demonstrated the expression of FasL on exosomes from term placentas as well. Interestingly, this study demonstrated the presence of FasL and TRAIL arranged in aggregates on exosome surfaces which can relay death signals to interacting cells including T cells in vitro [87]. Further skewing T cell function toward an inhibitory phenotype, EVs released from BeWo cells, a choriocarcinoma cell line used to model trophoblast cells, promote differentiation of helper T cells to T_regs_ and expansion of these anti-inflammatory cells through 10 kDa Heat Shock Protein expression [88].

While contributing to normal immunosuppression at the fetal-maternal interface in healthy pregnancy, EVs have also been implicated in complications such as preeclampsia (PE) (Figure 3). Clinically defined as the onset of hypertension with proteinuria during pregnancy, PE affects 2–8% of women worldwide every year and is a leading cause of maternal and fetal morbidity and mortality. While inflammation and defects in angiogenesis are described in models of PE, the cause(s) of these dysfunctions remains unknown [89]. Recent studies have demonstrated that EVs may have a role in the pathogenesis of PE. Increased levels of placenta-derived EVs are found in maternal circulation in women with early-onset or severe PE compared to healthy pregnant controls [90,91]. Placenta-derived EVs from PE remain in the circulation longer than placenta EVs from a healthy pregnancy. This is mediated by the increased expression of anti-phagocytic markers on the surface of PE placenta-derived EVs such as CD47 with concomitant downregulation of phosphatidylserine [92].

Treatment of trophoblasts with PE sera increases the release of large vesicles containing HMGB1 which is a signal of cell damage that induces sterile inflammation. Thus, these large particles can activate the endothelium to upregulate ICAM-1 potentially contributing to inflammation and leukocyte recruitment [93]. Trophoblasts from PE placentas secrete EVs which contain higher concentrations of miR-141. These EVs can induce T cell proliferation thus contributing to the shift in immunity toward a pro-inflammatory phenotype associated with PE [94]. Other miRNAs implicated in PE pathophysiology include miR-548c-5p. Downregulation of miR-548c-5p in serum EVs increases the secretion of IL-12 and TNF-α while increasing levels of nuclear NF-κB in macrophages. These inflammatory cytokines and activation of macrophages promote a shift in immunity to Th1 phenotype [95]. EVs released by STBs in PE can also activate inflammation. For example, STBs from PE placentas secrete exosomes which increase the production of superoxide by neutrophils. This effect may increase the formation of neutrophil extracellular traps (NETs) and is thought to contribute to the pathologic inflammation described in PE [96].

STB-derived EVs can also interact with monocytes and macrophages in PE. Larger EVs such as syncytial nuclear aggregates (SNAs or syncytial knots) and apoptotic bodies induce anti-inflammatory mediators. Specifically, co-culture of macrophages with SNAs promotes IL-10 secretion and IDO expression thus inhibiting inflammation. Downregulation of MHCII expression on macrophages is also induced and promotes decreased antigen presentation and T cell activation [97,98].

STB-derived EVs from PE placentas stimulate inflammatory cytokine production by PBMCs. Specifically, PBMCs secrete IL-1β in response to PE and first trimester EVs but not healthy, term EVs from STBs. PE STB-derived EVs also heighten cytokine responses to LPS indicating that EVs play a role in increasing baseline inflammation in PE [99]. A key feature of PE is the activation of platelets and a pro-coagulant state. EVs from PE placentas isolated by placental perfusion demonstrate high levels of tissue factor and can stimulate platelet activation. In this model treatment with aspirin can inhibit STB EV-induced platelet aggregation. This may provide insights into the efficacy of aspirin in the treatment of PE [100].

Other EV cargo proteins associated with inhibition of the immune system include syncytin-2 which is found on the surface of exosomes secreted by villous cytotrophoblasts (CTBs) and STBs. Encoded by an endogenous retrovirus, syncytin-2 is essential for syncytialization of CTBs to form STBs. Additionally, this protein has an immunosuppressive domain. Interestingly, levels of syncytin-2 are decreased in PE. The immunosuppressive domain of syncytin-2 decreases T cell activation and downregulates the production of Th1 cytokines. This study demonstrates that syncytin-2 on the surface of exosomes has the same effect when EVs are co-cultured with PBMCs. Knockdown of syncytin-2 in EV source cells ablated this effect demonstrating the role of EV cargo in the systemic immunologic shift from Th1 to Th2-dominant immunity in pregnancy [101].

In addition to their roles in immune system inhibition, EVs may also play a role in delivery by promoting the secretion of inflammatory cytokines such as IL-6 and IL-8 from maternal cells of the myometrium and decidua. These EVs also induced prostaglandin E2 (PGE_2_) secretion which promotes myometrial contractions [102].

## 4. Potent Clinical and Therapeutic Interest of EVs Derived from Ovarian Cancer or Placental Cells

### 4.1. EVs as Markers of Pathology

For both OC and PE, markers for early diagnosis are still not available. EVs represent an attractive diagnostic tool because they can be isolated from blood or other fluids offering patients a minimally invasive option compared to tissue biopsy. In addition, the lipid bilayer allows for EVs to package and protect their contents from enzymes found in the blood and tissues [103].

#### 4.1.1. EVs as Marker of OC

It has been demonstrated that ovarian tumor-derived EV contents differ from healthy tissue EVs, suggesting that they could be used in the diagnosis of OC or as a prognostic marker of overall survival [104]. Overexpression of several proteins has been identified in OC-derived exosomes compared to exosomes derived from healthy ovarian epithelial cells and may be useful in the diagnosis or prognosis of OC, for review [52,105]. Among these proteins, epithelial cell surface antigen (EpCAM), CD24, and CA125 have been confirmed by several different studies [106,107,108].

In addition to EV protein cargo, miRNA cargo may also be associated with clinical outcomes in OC. Several miRNAs have been identified as potential biomarkers for OC [109]. Specifically, comparison of miR-21, miR-141, miR200a, miR-200b, miR-200c, miR-203, miR-205, and miR-214 levels in serum-derived exosomes from healthy (10) and OC (50) patients demonstrated a significant difference in the level of miRNA expression. Furthermore, these miRNAs were stable in serum EVs. Together, these results suggest that exosome miRNA cargo could be used as clinical markers of disease [109].

In contrast, other EV miRNAs could be used as a marker of treatment response in OC. Analysis of six different miRNAs (let-7e, miR30c, miR-125b, miR-130a, miR-335) found in OC-derived exosomes demonstrated an upregulation of let-7e in paclitaxel-resistant OC cells and a downregulation in cisplatin resistant OC cells [110]. Conversely, miR-125b, miR-30c, miR-130a, and miR-335 were downregulated in all drug-resistant cell lines, suggesting that these miRNAs could be used as markers of treatment response in OC [110]. In addition to proteins and miRNA, several other molecules such as phosphatidylserine expressed in OC vesicles may be useful for OC diagnosis for review [111].

Despite these studies, further research is necessary to identify EV cargo with sufficient sensitivity and specificity for clinical use. Indeed, all current studies are conducted on small sample sizes. Nevertheless, EpCAM, CD24, and miR-200b could be promising markers, alone or in combination with other biomarkers, as they have been confirmed by several studies. Multicenter clinical trials are still required to validate the effectiveness of EV-based markers for OC diagnosis or prognosis.

#### 4.1.2. Placenta-Derived EVs as Markers of PE

PE is associated with placental dysfunction and altered circulating placental EVs in maternal plasma [90,91,112]. This observation has led several groups to investigate the diagnostic potential of placenta-derived EVs as a biomarker of PE (for review [113]).

These studies suggest that in addition to the observed altered level of placenta-derived EVs in maternal plasma [90,112,114], specific protein cargo (PLAP, Annexin V, TIMP-1, PAI-1, or glycosylated form of Siglec-6) [90,91,115,116] or miRNA content (in particular miR-486-1-5p and miR-486-2-5p) [90] is upregulated in plasma of women with PE and may be useful for diagnosis. However, the differences in isolation techniques of placenta-derived EVs, markers for EV subtype identification, and placenta-specific markers to verify the placental origin of EVs limit the development of relevant biomarkers for early diagnosis of PE.

### 4.2. Looking Ahead: Therapeutic Roles for EVs

The immune and angiogenesis modulation properties of EVs, the decreased risks associated with the transplantation of live cells and the lack of ethical concern make placental-derived EVs promising in regenerative medicine.

Salomon et al. observed that EVs released from placenta-derived MSC can induce migration and angiogenic capillary-like formation of placental microvascular endothelial cells, hence suggesting their possible therapeutic role in supporting vasculogenesis and angiogenesis for tissue repair in the ischemic setting [117].

Since this article, several studies have reported that placenta-derived EVs administration can suppress inflammatory response and/or promote cell regeneration in a different preclinical model of diseases such as pulmonary fibrosis [118], hepatic fibrosis [119]; vascular repair [120], kidney regeneration [121], lung repair [122], and Duchenne muscular dystrophy [123].

In addition to their own roles, EVs are also attractive candidates for drug delivery [124]. Indeed, EVs could be used to introduce miRNA to cancer cells to induce tumour suppressor genes. As proof of concept, exosomes were purified from primary-cultured omental fibroblasts of OC patients and loaded with a tumor suppressor (TS) miRNA [125]. Treatment with TS-miRNA-loaded-exosomes increased TS miRNA expression level in OC cell lines, suppressed the expression of TS-miRNA targets, and decreased cell proliferation and invasion. In xenograft mouse models, treatment with these exosomes also decreased peritoneal dissemination. Altogether, these studies suggest that miRNA replacement therapy using exosomes derived from non-malignant cells may be a promising tool for treatment of OC. 

Despite the great interest of EVs as a new therapeutic tool in regenerative medicine and cancer treatment, there are still challenging issues regarding their use. These include the development of a standardization method for their isolation, storage and administration route, and the limited information about their effects on tissue behavior. 

## 5. Conclusions and Future Perspective

Many studies have demonstrated the crucial role of EVs in ovarian cancer and pregnancy. Trophoblast and OC-derived EVs are involved in cell invasion, migration, angiogenesis, and immune modulation. However, if the functions of OC and trophoblast-derived EVs are similar, their mechanism of action is different.

EVs can provide many opportunities for the development of clinical tests for the diagnosis and prognosis of OC or PE, or therapeutic tools in regenerative medicine and cancer treatment. However, many limitations hinder the proper study of EVs as well as their use as a diagnostic or therapeutic tool. Specifically, the lack of a standardized protocol for EV isolation prevents consistent comparisons between studies. Many protocols are used to isolate EVs including centrifugation, precipitation, chromatography-based methods, immunological separation techniques and microfluidic devices. Different methods create inconsistencies between studies and prevent rigorous comparison and accurate interpretation of the current literature. Thus, a standardization of the methods used to isolate EVs is necessary to establish standards for diagnostic testing. Additionally, an improvement in the methods to characterize EVs is necessary. Since EVs of different sizes and or different markers have different functions, it is imperative to know which specific type is being analyzed for diagnosis or used for therapy. Many technologies are used to characterize EVs including ELISA, flow cytometry, dynamic light scattering, nanoparticles tracking analysis, and new technologies such as nano-plasmonic exosome (nPLEX) assay, integrated magneto-electrochemical exosome (iMEX), ExoCounter, ExoSearch, and Microfluidic affinity separation chip. The development of these last micro- and nano-technologies increased the sensitivity of EV characterization and isolation and may overcome the technical challenges limiting the clinical potential of these vesicles in the near future.

## Figures and Tables

**Figure 1 biomedicines-09-01257-f001:**
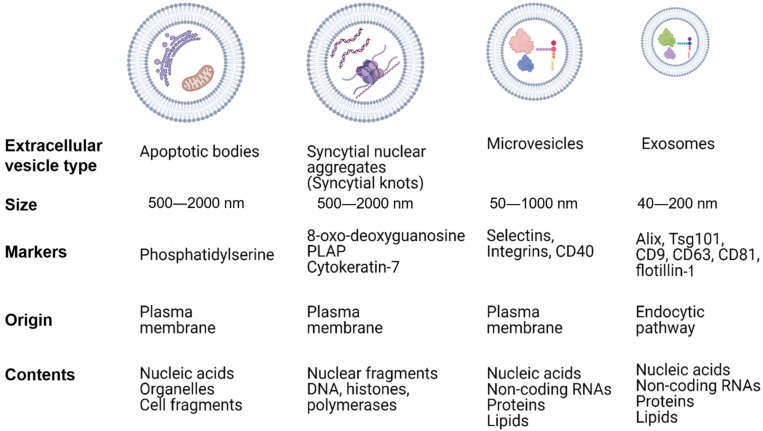
Characteristics of extracellular vesicles based on size, surface marker expression, origin, and cargo.

**Figure 2 biomedicines-09-01257-f002:**
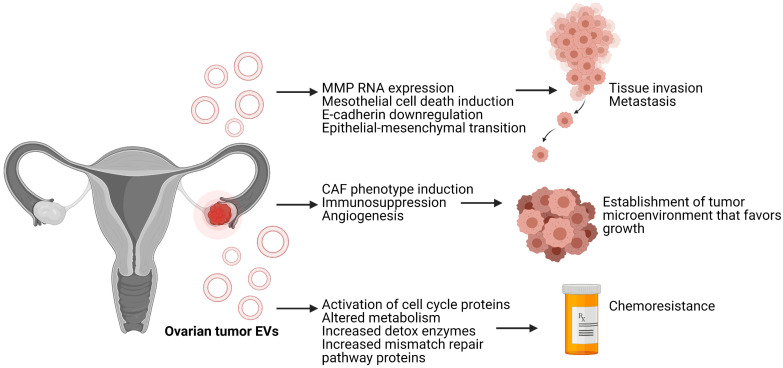
Mechanisms of invasion, growth, and chemoresistance in ovarian cancer mediated by tumor-derived extracellular vesicles.

**Figure 3 biomedicines-09-01257-f003:**
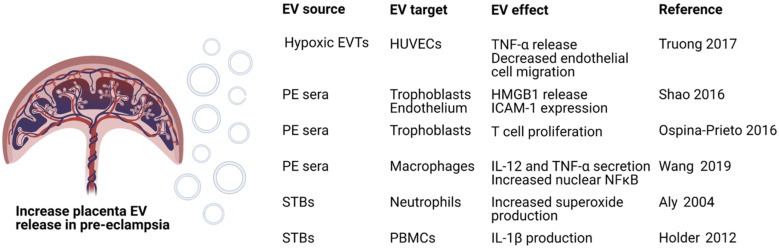
Effects of EVs on different cell types in PE.

## Data Availability

Not applicable.

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
