# Peer review of "The Role and Clinical Interest of Extracellular Vesicles in Pregnancy and Ovarian Cancer"

_biomedicines, 2021, doi:10.3390/biomedicines9091257_

Round 1
Reviewer 1 Report
The review article by Kazemi et al., entitled “The role and clinical utility of extracellular vesicles in pregnancy and ovarian cancer” discussed and summarized the role of EVs in ovarian cancer metastasis, chemoresistance, and immune modulation. The article also discusses the clinical utility of EVs as markers and therapeutics for ovarian cancer and pre-eclampsia. Overall the manuscript is well written and composed. The introduction is informative and contains useful information related to the study presented. Critical and systematic reviews like this would help researchers in the field to get updated information at one place and the article also falls under the scope of the journal Biomedicines.
The transitions from paragraph to paragraph are well connected and don’t feel fragmentary and the logical rhythm is also maintained. I have observed only a couple of informational errors (see comments). Therefore, the manuscript can be considered for publication once the suggestions are addressed and also based on comments from other referees.
- Line 101-102: Apoptotic bodies express Annexin V and phosphatidylserine on their surfaces [34]…It’s an established fact that apoptotic bodies express phosphatidylserine (PS) on their outer surfaces which in turn mark the cell for phagocytosis, but I’m not very sure about the expression of Annexin V on the surface of apoptotic bodies. Annexin V binds to PS with high affinity and specificity and therefore used in many assays to identify apoptotic cells. Provide additional reference in support of this.
- Line 108: Lastly, exosomes are the smallest EVs ranging in size from 40-120 nm in diameter… Many researchers observed that the size of exosomes greatly varies and it usually ranges from 40-200nm in diameter. I would recommend using this size range including in the fig 1.
- Line 115: replace with flotillin-1.
Author Response
Dear reviewer,
we would like to thank you for your time reviewing this article and your valuable comments that improved the quality of our manuscript. We have modified our paper according your recommendations.
- We removed Annexin V as marker of apoptotic bodies (in the text and in the figure 1)
- We modified the size range of exosomes (in the text and in the figure 1)
- We replaced fotillin ba flotillin-1 (in the text and in the figure 1)
Reviewer 2 Report
The manuscript authored by Kazemi et al summarized the research progress in the role and clinical potential of extracellular vesicles in pregnancy and ovarian cancer. It is a very interesting topic, covering two different but relevant fields, pregnancy and ovarian cancer. While the manuscript is easily understood, it can be improved in a number of aspects
- Role of exosome in ovarian cancer and pregnancy have been separately reviewed quite often, but it is important to discuss them side-by-side, which may lead to new insights into the function of exosome in two biological processes. The Introduction of the manuscript is very enjoyable to read, with the significance of exosome in ovarian cancer and pregnancy discussed together. In the later sections, however, ovarian cancer and pregnancy are talked about completely independent to each other. It’d be better if the authors can detail the exosome-involved pathways or cellular procedures shared in pregnancy and ovarian cancer.
- A future remarks section is expected. Section 4.2 doesn’t focus on this, a part of Conclusions actually talks about future perspectives.
Minor points
- Exosome as a biomarker or treatment option hasn’t been used in clinical practice, it is inappropriate to use “clinical utility” in the title
- Page 4, classification of ovarian cancer is incorrect, mucinous subtype is of epithelial origin.
- Page 4, sentence “Seeding of the peritoneum and spread through ascitic fluid occurs early in disease and contributes to the poor prognosis associated even with early stages of ovarian cancer” needs to be reworded. Stage I ovarian cancer is limited to ovary, with no peritoneal spreading, and the prognosis isn’t too bad, with >75% of patients survive beyond 5 years.
- Page 4, sentence “Recent studies demonstrate that one mechanism by which malignant cells from ovarian tumors can gain access to the ascitic fluid is through EVs” overstates the role of EV in OC peritoneal spreading. EV-induced degradation of extracellular matrix and apoptosis of mesothelium surely facilitates the escaping of ovarian cancer cells, but won’t be a driving mechanism.
- Page 4 sentence “Once the peritoneum is involved, tumor cells have access to abdominal organs and can spread further especially with involvement of the liver.” is inaccurate. Peritoneum describes a layer of membrane that lines the abdominal cavity, there is no evidence supporting the metastasis order the authors described.
Author Response
Dear reviewer,
we would like to thank you for your valuable comments that improved the quality of our manuscript.
We have modified our paper according to your recommendations.
- "Role of exosomes in ovarian cancer and pregnancy have been separetely reviewed quite often." The functions of extracellular vesicles in pregnancy and ovarian cancer are quite similar. However, their mechanism of action is different. That is why we discussed the significance of extracellular vesicles in pregnancy and ovarian cancer together and we then separated their mechanism of action for clarity and fluidity in reading.
- "A future remarks section is expected". We replaced the conclusion section by a section entitled "Conclusion and future perspective".
- We replaced "clinical and therapeutic application of EVs" by "potent clinical and therapeutic interest of EVs"
- We modified the classification of ovarian cancer and focused on epithelial ovarian cancer
- The sentence has been modified. The term "early in the disease" was removed
- We agree with the reviewer that we overstate the role of EVs in OC peritoneal spreading. Our sentence was replaced by "EVs released from ovarian cancer cells may facilitate the escaping of ovarian cancer cells"
- This sentence was completely removed.